# Effects of Lipid Lowering Therapy Optimization by PCSK9 Inhibitors on Circulating CD34+ Cells and Pulse Wave Velocity in Familial Hypercholesterolemia Subjects without Atherosclerotic Cardiovascular Disease: Real-World Data from Two Lipid Units

**DOI:** 10.3390/biomedicines10071715

**Published:** 2022-07-15

**Authors:** Roberto Scicali, Giuseppe Mandraffino, Michele Scuruchi, Alberto Lo Gullo, Antonino Di Pino, Viviana Ferrara, Carmela Morace, Caterina Oriana Aragona, Giovanni Squadrito, Francesco Purrello, Salvatore Piro

**Affiliations:** 1Department of Clinical and Experimental Medicine, University of Catania, 95100 Catania, Italy; nino_dipino@hotmail.com (A.D.P.); vivi.fer@hotmail.it (V.F.); francesco.purrello@unict.it (F.P.); spiro@unict.it (S.P.); 2Department of Clinical and Experimental Medicine, University of Messina, University Hospital G. Martino, Lipid Center, 98100 Messina, Italy; mscuruchi@unime.it (M.S.); carmela.morace@unime.it (C.M.); oriana.aragona@hotmail.it (C.O.A.); gsquadrito@unime.it (G.S.); 3Unit of Rheumatology, Department of Medicine, ARNAS Garibaldi Hospital, 95100 Catania, Italy; albertologullo@virgilio.it

**Keywords:** circulating CD34+ cells, familial hypercholesterolemia, PCSK9 inhibitors, pulse wave velocity, cardiovascular risk

## Abstract

Background: Circulating CD34+ progenitor cells (CD34+CPCs) are characterized by pronounced tissue regeneration activity. Dyslipidemic subjects seemed to have reduced CD34+CPCs, and statin therapy appeared to restore their levels. We aimed to evaluate the effects of PCSK9 inhibitors (PCSK9-i) on CD34+CPCs and pulse wave velocity (PWV) in a cohort of heterozygous familial hypercholesterolemia (HeFH) subjects. Methods: We determined CD34+ cell count and its change after PCSK9-i in 30 selected HeFH subjects and 30 healthy controls. Lipid profile and PWV were evaluated at baseline (T0), 6 months after intensive lipid lowering strategy (statin plus ezetimibe, T1), and after 6 months of optimized therapy with PCSK9-i (T2); CD34+ cell count was reported at T1 and T2. Results: At T1, the median value of CD34+ cells was not significantly different between HeFH subjects and controls, and the same result was obtained at T2. PWV was significantly reduced at T1 (ΔPWV − 14.8%, *p* < 0.001 vs. T0) and T2 (ΔPWV − 10.96%, *p* < 0.001 vs. T1). Dividing HeFH subjects into two groups of high- and low-CD34+ cell count, CD34+CPCs appeared to be polarized with a significant difference between the two groups (1.2 (0.46) vs. 4.74 (1.92), *p* < 0.001), also with respect to controls (both *p* < 0.001). This polarization was no longer observed at T2, and neither with respect to controls. ΔCD34+ was +67.4% in the low-CD34+ group and −39.24% in the high-CD34+ group (*p* < 0.001). Lastly, we found a significant correlation between ΔCD34+ cell number and ΔPWV in HeFH subjects (rho = −0.365, *p* < 0.05), particularly in the low-CD34+ group (rho = −0.681, *p* < 0.001). Conclusion: PCSK9-i exhibited favorable effects on CD34 + CPCs as was on PWV values in a cohort of FH subjects. Our preliminary findings suggest a possible positive role of this novel lipid-lowering strategy on vascular homeostasis.

## 1. Introduction

Endothelial dysfunction is considered the *primum movens* of the atherosclerotic process, and its progression leads to atherosclerotic cardiovascular disease (ASCVD) [1,2]. An increased LDL-C plasma level promotes the accumulation of lipid-laden macrophages in the arterial wall that favors the progression of endothelial dysfunction and thus atherosclerotic injury [3,4]. Beyond the arterial lipid storage, an impaired balance between endothelial damage and repair is crucial for atherosclerosis progression [5].

Circulating progenitor cells (CPCs) are derived from bone marrow stem cells and they are characterized by pronounced tissue regeneration activity [6]. In particular, CD34+CPCs may contribute to vascular health by promoting endothelial turnover and angiogenesis [7]; thus, CD34+CPCs could play an important role in endothelial repair that is crucial for contrasting atherosclerosis progression. CD34+CPCs could be inversely modulated by cardioinflammatory status, and a low CD34+CPC number was associated with an increased risk of death in subjects with coronary artery disease [8,9]. In this context, CD34+CPCs could be a novel cardiovascular biomarker able to identify subjects with higher cardiovascular risk. Dyslipidemic subjects seemed to have a reduced CD34+ cell number, and statin therapy appeared to restore their levels [10,11]. 

The inhibitors of proprotein convertase subtilisin/kexin type 9 (PCSK9-i) are the cornerstone of novel lipid-lowering therapies (LLT) [12]. The main role of PCSK9-i is to bind the circulating PCSK9; thus, a reduction in LDL-C receptor cleavage and an increase in its presence on the liver cell surface were observed [13]. The clinical efficacy of PCSK9-i was shown in previous studies [14,15]; in particular, the reduction in LDL-C and ASCVD by PCSK9-i was ≅50–60% and 15%, respectively. Because of its lipid-lowering and cardiovascular properties, PCSK9-i therapy is crucial in subjects with a monogenic cause of high LDL-C such as familial hypercholesterolemia (FH) [16]. No data exist regarding the effect of PCSK9-i on CD34+CPCs in this population. 

In this study, we aimed to evaluate the effects of PCSK9-i on CD34+CPCs and on early atherosclerosis damage, evaluated by pulse wave velocity (PWV) in a cohort of FH subjects.

## 2. Methods

### 2.1. Study Design and Population

This was a retrospective, observational study involving patients with a previously confirmed FH genetic diagnosis [17]. 

Patients included in this study were selected from a previous prospective, observational study involving a larger cohort of heterozygous FH (HeFH) subjects [18], involving two lipid centers; thus, in this study, we performed an additional analysis on CD34+ cell count and its change after PCSK9-i in 30 selected HeFH subjects and 30 healthy controls. All participants were enrolled from the Lipid Centre of the University Hospital of Messina, and the University Hospital of Catania, Italy, from September 2017 to May 2019. The detailed inclusion and exclusion criteria were already defined in the previous study. However, in this study, we did not include patients with ASCVD, or those already diagnosed with type 2 diabetes (T2D) or arterial hypertension.

Briefly, physical examination and clinical history were recorded at the time of enrolment. After a 12 h fast, all patients performed the assessment of standard hematological and clinical biochemistry parameters, and PWV evaluation. Body weight and height were measured, and the body mass index was calculated (kg/m^2^). From our database, we selected 30 HeFH patients with a baseline lipid profile and PWV, lipid profile, PWV, and CD34+ cell count at T1 (after 6 months of the maximally tolerated statin plus ezetimibe treatment) and T2 (after 6 months of PCSK9-i add-on therapy), and familial, clinical, and pharmacological histories. Thirty healthy subjects matched by age served as controls for the circulating CD34+ cell number.

The study was approved by the local ethics committee (prot. number 46/19) in accordance with the ethical standards of the institutional and national research committees, and with the 1964 Declaration of Helsinki and its later amendments or comparable ethical standards. Informed consent was obtained from each subject enrolled in the study.

### 2.2. Biochemical Analysis

Serum total cholesterol (TC), triglycerides (TGs), and high-density lipoprotein cholesterol (HDL-C) were assessed with the available enzymatic methods [18]. LDL-C was obtained with the Friedewald formula, and non-HDL-C was derived from baseline values. 

### 2.3. CD34+ Cell Count

Circulating CD34+ cells were determined by means of flow cytometry; following ISHAGE recommendations, we chose to determine the absolute count instead of estimating it through frequencies [19]. The technique was explained in detail elsewhere [20]. Briefly, 50 µL of peripheral blood was incubated with 10 µL of PE-conjugated antihuman CD34 antibody (BD) in TRUCOUNT tubes (BD) for 15 min. Sample acquisition and analysis were performed using a FACSCalibur cytometer with CELLQuest software. Nonviable cells were excluded according to 7-amino-actinomycin D (7-AAD; BD Pharmingen, San Diego, CA, USA) staining. Circulating cells with stem cell antigen CD34 were defined as progenitor hematopoietic CD34+ cells, and they were estimated and counted (cells/mL) as an absolute count as previously described.

### 2.4. Pulse Wave Velocity Evaluation

PWV analysis was performed as previously described [18]; briefly, the SphygmoCor CVMS (AtCor Medical, Sydney, Australia) the system used for PWV evaluation was composed of a tonometer and 2 different pressure waves recorded at the common carotid artery (proximal recording site) and at the femoral artery (distal recording site). An electrocardiogram was performed to determine the start of the pulse wave. The PWV was defined as the difference in the time interval of the pulse wave between the 2 different recording sites and the heart, divided by the travel distance of the pulse waveform, and it was calculated on the mean of 10 consecutive pressure waveforms to perform a complete respiratory cycle.

### 2.5. Statistical Analysis

Since CD34+ cell numbers were not normally distributed, as verified by the Kolmogorov–Smirnov test, a classical nonparametric approach was chosen. Data are presented as median and interquartile range (IQR) for continuous variables, and as frequencies or percentages where appropriate. The difference between the different timepoints was tested with the Kruskal–Wallis test, and the Wilcoxon test was carried out to verify the differences in CD34+ cell number, PWV, and lipid parameters between T1 and T0 (where available), T2 and T1, and between T2 and T0 (where available).

We selected 30 healthy subjects from our database with an SPSS case–control matching function to obtain an age- and sex-matched control population. Comparisons of the study variables between subjects and controls were carried out with the Mann–Whitney test.

Changes in TC, HDL-C, TG, LDL-C, CD34+ cells and PWV, from baseline (T0) for both the treatment times (T1, T2), were evaluated as delta (∆) and calculated according to the following formulas: [((T1 − T0)/T0)*100)) and ((T2 − T1)/T1)*100))].

Once the distribution of CD34+ cells in controls and in HeFH was assessed at the first time point, in the secondary analyses, we divided the study population into two groups (high-CD34+ and low-CD34+) on the basis of the median CD34+ cell value of the control subjects. Statistical analyses were then repeated within HeFH at the two time points.

Moreover, as a preliminary exploration, we verified the potential inter-relationships between the changes in CD34+ cell number change with the PWV change after six months of PCSK9-i (T2 − T1, Δ) with Spearman’s test. 

A value of *p* < 0.05 was chosen to denote statistical difference.

## 3. Results

Table 1 shows the main characteristics of 30 HeFH subjects and 30 controls; plasma lipids are reported at baseline (T0), after 6 months of intensive lipid lowering strategy with statin plus ezetimibe (T1), and after 6 months of optimized therapy with PCSK9-i (T2); CD34+ cell count is reported for T1 and T2. 

Concerning cardiovascular risk factor, blood pressure and LDL-C values were higher in HeFH subjects than those in the controls (SBP 135 (10) vs. 120 (20) mmHg, *p* < 0.001; DBP 80 (13) vs. 70 (10) mmHg, *p* < 0.001; LDL-C 299 (98) vs. 96 (35.5) mg/dL, *p* < 0.001). 

Regarding statin therapy, at the first time point, the majority of patients were on 20 mg rosuvastatin (12 patients), followed by 10 mg rosuvastatin (10 patients), 40 mg atorvastatin (4 patients), and 20 mg atorvastatin (4 patients). All patients were also recommended to take 10 mg ezetimibe per day. LDL-C was 163.9 ± 62.8 mg/dL under the consolidated prescription of the maximal tolerated statin-plus-ezetimibe LLT (−46.2%, *p* < 0.001 vs. baseline; *p* < 0.001 vs. controls). 

At the second time point, PCSK9-i therapy was started; more specifically, in 50%, 140 mg evolocumab was prescribed (15 patients) and in 50%, 150 mg alirocumab (15 patients). After six months of PCSK9-i add-on therapy, LDL-C was 74.8 ± 38.8 mg/dL (−52.9%, *p* < 0.001 vs. T1) and significantly lower than that of the controls (*p* < 0.001) (Table 2).

After six months of statin and ezetimibe, the median value of CD34+ cell number was similar between HeFH and controls (HeFH 1.77 (3.55) cells/μL, controls 2.12 (1.95) cells/μL; *p* = 0.639) (Table 3). However, the very wide IQR reported for the value distribution in HeFH suggested a potential polarization in HeFH; therefore, we divided the HeFH population into two groups according to the CD34+ cell median value of control subjects (19 subjects, low-CD34+ HeFH group, 11 subjects, high-CD34+ HeFH group). The patients of the low-CD34+ HeFH group were older than those of the high-CD34+ HeFH group (61 (11) vs. 47 (14) years, *p* = 0.002). Although at T1, the CD34+ cell number was similar in the whole HeFH group and controls, and the CD34+ cell number in the whole HeFH group at T2 vs. T1 values (2.2 (1.95). vs. 1.77 (3.55), *p* = 0.709), after division into high- and low- CD34+ cell count groups, we confirmed the polarization of the T1 CD34+ cell values with a significant difference between the two HeFH groups (1.2 (0.46) vs. 4.74 (1.92), *p* < 0.001) and also as compared to controls (for both *p* < 0.001). At T2, the polarization of CD34+ cells was not observed; in fact, we found no difference between the two HeFH groups (2.1 (0.43) vs. 2.45 (0.83) cells μ/L; *p* = 0.164) and no difference between HeFH groups and controls (high-CD34+ vs. controls, *p* = 0.952; low-CD34+ vs. controls, *p* = 0.419). However, ΔCD34+ was different between the two HeFH groups: +67.4% in the low-CD34+ group vs. −39.24% in the high-CD34+ group (for both, *p* < 0.001) (Table 4, Figure 1).

HeFH subjects exhibited a higher baseline PWV than the controls did (10.72 (3.7) vs. 4.9 (0.75) m/s, *p* < 0.001); at T1, PWV decreased to 8.71 (2.6) m/s, (ΔPWV − 14.8%, *p* < 0.001 vs. T0), while at T2 it was 7.66 (2.1) m/s (ΔPWV − 10.96%, *p* < 0.001 vs. T1) (Table 3, Figure 2).

At baseline, PWV was 11.6 (3.9) and 8.9 (3.4) in the low and high-CD34+ groups, respectively (*p* = 0.075). However, PWV change was different in the high-CD34+ and low-CD34+ groups. PWV decreased to 9.45 (2.1) at T1 (ΔPWV − 15.5%, *p* < 0.001 vs. T0) and to 8.32 (1.9) at T2 (ΔPWV − 11.9%, *p* < 0.001 vs. T0) in the low-CD34+ group, while in the high-CD34+ group, it decreased to 7.4 (2.3) at T1 (ΔPWV − 13.5%, *p* < 0.001 vs. T0) and to 6.9 (1.6) at T2 (ΔPWV − 9.2%, *p* < 0.001 vs. T1) (*p* < 0.001). As verified by the MWW test, while the ΔPWV was comparable in the two HeFH groups at T1 (*p* = 0.491), at T2 the low-CD34+ group experienced a significantly wider improvement, at least in terms of percentage (*p* = 0.034) (Table 4). 

As preliminary analysis, we also verified the potential inter-relationships between CD34+ cell number and PWV changes after six months of therapy with PCSK9-I; we found a significant correlation between ΔCD34+ cell number and ΔPWV in the HeFH population (rho = −0.365, *p* < 0.05); as well as in the low-CD34+ HeFH group (rho = −0.681, *p* < 0.001), while in the high-CD34+ HeFH group the Δ changes appeared to not be correlated (rho = 0.217, *p* = 0.356). 

## 4. Discussion

In the last decade, the field of cardiovascular research has focused on the discovery of novel circulating biomarkers as key targets of innovative cardiovascular therapies [21]; in this context, CD34+ cell number could be considered to be a promising candidate in the cardiovascular field. In this study, we evaluated the effect of lipid-lowering therapy optimization by PCSK9-i on CD34+ cell number and pulse wave velocity in FH subjects; to the best of our knowledge, this is the first study evaluating the role of PCSK9-i on endothelial homeostasis in this population. Six-month PCSK9-i therapy was able to significantly change CD34+ cell behavior, increasing the number in FH subjects with a low CD34+ cell count, and lowering it in FH subjects with a high CD34+ cell count. Moreover, PCSK9-i significantly improved mechanical artery properties assessed through PWV in HeFH subjects. Lastly, ΔCD34+CPCs appeared to be correlated with ΔPWV in the whole population and particularly in the low-CD34+ group. 

The balance of circulating CD34+ cells could represent an intriguing regulator of vascular homeostasis [22]. Previous studies described a bimodal action of circulating CD34+ cell number in atherosclerosis development [23,24]. In fact, Fadini et al. showed that a low circulating CD34+ cell number predicted cardiovascular events and death from any cause in a cohort of dysmetabolic subjects; however, Mandraffino et al. found in a cohort of hypertensive subjects that the CD34+ cell number was increased in subjects with low cardiovascular damage, but it was reduced if an advanced cardiovascular injury was present. Circulating CD34+ cells are derived from bone marrow hematopoietic progenitor cells, and they were proposed to differentiate in several different cell types involved in vascular homeostasis [25] and into foam cells [26]. 

A possible explanation of the different maturation of CD34+ cells could be the relationship with the inflammatory status [27,28]. In fact, in a cohort of subjects with rheumatoid arthritis, Lo Gullo et al. found that an imbalance of the redox system was associated with increased cell senescence and apoptosis, leading to a reduced circulating CD34+ cell number. On the other hand, Mandraffino et al. showed that redox homeostasis could increase the number of CD34+ cells in a cohort of uncomplicated hypertensive subjects. However, Shimizu et al. recently reported that a higher circulating CD34+ cell number was associated with arterial wall thickening in a cohort of elderly subjects without hypertension [29]; thus, an early beneficial abundance of circulating CD34+ cells could paradoxically become a deleterious resource for atherosclerosis development. In line with these findings promoting the concept of *in medio stat virtus*, FH subjects with low or high CD34+ cell numbers here had a higher PWV compared to controls. Baseline PWV was higher in FH than that in the controls, and this could have been influenced by the higher-baseline, long-lasting levels of LDL-C and SBP in FH compared to those in the controls. 

Regarding the role of PCSK9 on bone-marrow-derived hematopoietic progenitor cells, recent experimental and clinical studies reported a negative effect of PCSK9 on circulating CD34+ cell homeostasis [30,31,32]. In fact, in cultured human aortic vascular smooth muscle cells (SMCs), Guo et al. showed that PCSK9 induced the senescence and apoptosis of CD34+ cell-derived SMCs, thus promoting vascular injury progression; furthermore, Tripaldi et al. showed that PCSK9 plasma levels are inversely correlated with the number of circulating CD34+ cells in a cohort of diabetic subjects, and Chao et al. showed that circulating PCSK9 are positively associated with apoptotic circulating endothelial cell number in atherosclerotic subjects. Schuster et al. showed that the number of circulating CD34+ cells significantly increased after the inhibition of PCSK9 in an APOE*3Leiden. CETP mouse model [33]. Considering these findings, PCSK9 inhibition seemed to be able to contrast the negative effect of PCSK9 on circulating CD34+ cells. In line with this hypothesis, in our study, we found that six-month PCSK9-I therapy significantly increased the number of circulating CD34+ cells in FH subjects with a low CD34+ cell count, and it reduced the circulating CD34+ cell number in FH subjects with a higher CD34+ cell level; moreover, in both FH groups, the obtained CD34+ cell numbers were similar to those of the controls. Furthermore, PCSK9-i treatment promoted a favorable change of circulating CD34+ cell amount and it significantly reduced PWV values in FH subjects; thus, this lipid-lowering strategy appeared to limit atherosclerotic injury progression in this population. 

There are several limitations to our study; first, this was a retrospective, observational study, and the PCSK9-i therapeutic strategy thus depended on the physician’s decision. Furthermore, other parameters that may explain the interaction of CD34+ cell number and endothelial dysfunction, such as flow-mediated dilation, redox status, and oxidative stress determination, were not evaluated. Moreover, lipoprotein(a) (Lp(a)) levels were not evaluated, while previous studies showed that PCSK9-i therapy was able to reduce Lp(a) levels in a large cohort of subjects [34]. The study population size was small; however, we demonstrated significant changes of CD34+ cell number and PWV values after PCSK9-i therapy in FH subjects. These preliminary findings should be confirmed in a larger study population through defined diagnostic tools and statistical analyses. 

## 5. Conclusions

PCSK9-i treatment significantly changed the number of CD34+ cells and PWV values in a favorable way in a cohort of FH subjects. Our preliminary findings suggest a possible positive role of this novel lipid-lowering strategy on vascular homeostasis; however, a randomized controlled prospective trial is required to evaluate the effect of PCSK9-i therapy on these pathways in a large cohort of FH subjects.

## Figures and Tables

**Figure 1 biomedicines-10-01715-f001:**
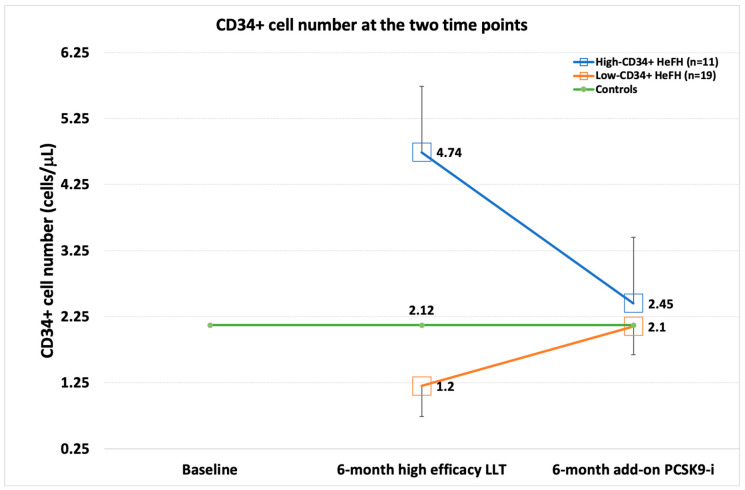
CD34+ cell number at the two scheduled time points in the low-CD34+ cell group and the high-CD34+ cell group; IQR is represented by error bars. Green line represents the median value of baseline CD34+ cells in controls.

**Figure 2 biomedicines-10-01715-f002:**
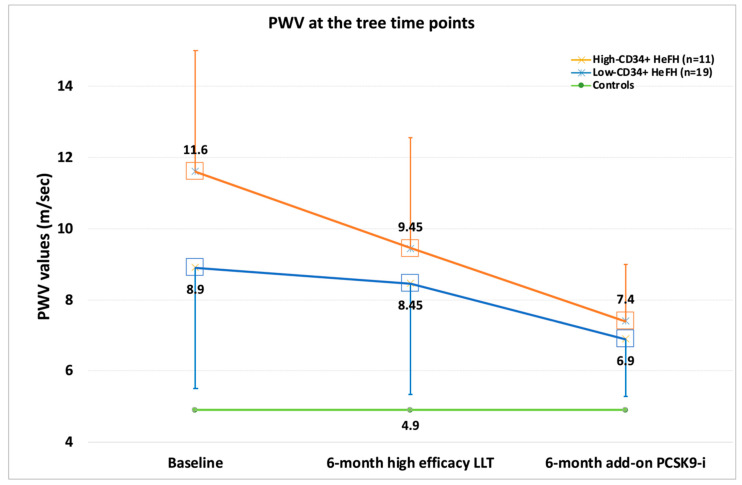
PWV values at the three scheduled time points in the low-CD34+ cell group and the high-CD34+ cell group; IQR is represented by error bars. Green line represents baseline PWV median value in controls.

**Table 1 biomedicines-10-01715-t001:** Characteristics of the study population.

	HeFH	Controls	*p*
N	30	30	
Age, years	54.5 (17)	52 (13)	0.456
Men, n [%]	17 [56.7]	13 [43.3]	0.302
BMI Kg/m^2^	26.4 (7.7)	24 (3.7)	0.09
SBP mmHg	135 (10)	120 (20)	<0.05
DBP mmHg	80 (13)	70 (10)	<0.05
PWV m/s	10.7 (3.7)	4.9 (0.75)	<0.001
**FH genotype**			
LDLR, n [%]	30 [100]		
**Mutation class**			
Amino acid change, n [%]	18 [60.0]		
Null allele, n [%]	12 [40.0]		
**FH phenotype**			
Heterozygous FH, n [%]	30 [100]		
**Statin therapy**			
Rosuvastatin 20 mg, n [%]	12 [40]		
Rosuvastatin 10 mg, n [%]	10 [33.3]		
Atorvastatin 40 mg, n [%]	4 [13.3]		
Atorvastatin 20 mg, n [%]	4 [13.3]		
Ezetimibe 10 mg n [%]	30 [100]		

Data are presented as median (interquartile range) or numbers and percentages [%]. Abbreviations: BMI: body mass index; SBP: systolic blood pressure; DBP: diastolic blood pressure; PWV: pulse wave velocity. *p* = significance levels for Mann–Whitney test (continuous variables) or chi-squared test (categorical variables). Round brackets for IQR, square brackets for percentage.

**Table 2 biomedicines-10-01715-t002:** Main HeFH subject characteristics over time points.

	HeFH Subjects (*n* = 30)	HeFH Subjects (*n* = 30)	HeFH Subjects (*n* = 30)	∆ T1-T0 (*p*)	∆ T2-T1 (*p*)
	Baseline	Six-Month Statin + Ezetimibe	Six-Month Add-on PCSK9-i		
**Risk factors**					
Hypertension	0	0	0	-	-
Type 2 diabetes, n	0	0	0	-	-
ASCVD	0	0	0	-	-
**Lipid Profile**					
TC, mg/dL	350 (80)	222.5 (85)	140.5 (64)	<0.001	<0.001
HDL, mg/dL	50.5 (19)	49.5 (22)	51 (19)	0.861	0.512
TG, mg/dL	98.5 (52)	105.5 (83)	88 (40)	0.212	0.109
LDL-C, mg/dL	299 (98)	149.3 (65.5)	66.1 (49)	<0.001	<0.001
**CV risk markers**					
CD34+ cells (cells/μL)	-	1.7 (3.55)	2.2 (1.7)	-	0.709
PWV (m/s)	10.72 (3.7)	8.71 (2.6)	7.66 (2.1)	<0.001	<0.001

Comparisons between two consecutive time points were carried out by Wilcoxon test (*p*).

**Table 3 biomedicines-10-01715-t003:** Analysis of change in CD34+ cell number and PWV over time points in HeFH subjects.

		CD34+ Cells		(*p*)	∆		(*p*)
**Controls**		2.12 (1.95)					
**HeFH**	**T0**	-	T0 vs. ctrls	-			
	**T1**	1.77 (3.55)	T1 vs. ctrls	=0.639	-	-	-
	**T2**	2.2 (0.7)	T2 vs. ctrls	=0.685	+22.57%	T2 vs. T1	=0.709
		**PWV**		**(*p*)**	**∆**		** *(p)* **
**Controls**		4.9 (0.75)					
	**T0**	10.7 (3.7)	T0 vs. ctrls	<0.001			
**HeFH**	**T1**	8.7 (2.5)	T1 vs. ctrls	<0.001	−14.80%	T1 vs. T0	<0.001
	**T2**	7.6 (1.9)	T2 vs. ctrls	<0.001	−10.95%	T2 vs. T1	<0.001

ctrls: controls.

**Table 4 biomedicines-10-01715-t004:** Analysis of change in CD34+ cell number and PWV over time points in HeFH subjects as subdivided according to baseline CD34+ values.

		CD34+ Cells		(*p*)	∆	(*p*)	(*p*) Low vs. High
**Controls**		2.12 (1.95)					
**L-CD34+**	**T0**	-	T0 vs. ctrls	-			
	**T1**	1.2 (0.46)	T1 vs. ctrls	<0.001	-	-	
	**T2**	2.1 (0.43)	T2 vs. ctrls	0.952	+67.45%	<0.001 vs. T1	
**H-CD34+**	**T0**	-	T0 vs. ctrls	-			
	**T1**	4.74 (1.92)	T1 vs. ctrls	<0.001	-	-	*p* < 0.001
	**T2**	2.45 (0.83)	T2 vs. ctrls	0.419	−39.24%	=0.008 vs. T1	*p* = 0.164
		**PWV**		**(*p*)**	**∆**		**(*p*) low vs. high**
**Controls**		4.9 (0.75)					
	**T0**	11.6 (3.9)	T0 vs. ctrls	<0.001			
**L-CD34+**	**T1**	9.45 (2.1)	T1 vs. ctrls	<0.001	−15.5%	<0.001 vs. T0	
	**T2**	8.32 (1.9)	T2 vs. ctrls	<0.001	−11.9%	<0.001 vs. T1	
	**T0**	8.9 (3.4))	T0 vs. ctrls	<0.001			=0.075
**H-CD34+**	**T1**	7.4 (2.3)	T1 vs. ctrls	<0.001	−13.5%	<0.001 vs. T0	<0.05
	**T2**	6.9 (1.6)	T2 vs. ctrls	<0.001	−9.2%	<0.001 vs. T1	=0.055

CD34+ cell number is reported as cells/mL; PWV as m/s; ctrls: controls. L-CD34+, H-CD34+: low and high CD34+ cell groups. (*p*) Tx vs. ctrls: significance levels for Mann–Whitney test; ∆: percentage change between two consecutive time points; *(p*) vs. Tx: significance levels for ∆ change between two consecutive time points, Wilcoxon test. *(p)* low vs. high: comparisons of CD34+ cell number or PWV values between high- vs. low- CD34+ groups, Mann–Whitney test.

## Data Availability

No publicly archived datasets were used.

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
