# Peer review of "Effects of Lipid Lowering Therapy Optimization by PCSK9 Inhibitors on Circulating CD34+ Cells and Pulse Wave Velocity in Familial Hypercholesterolemia Subjects without Atherosclerotic Cardiovascular Disease: Real-World Data from Two Lipid Units"

_biomedicines, 2022, doi:10.3390/biomedicines10071715_

Round 1

Reviewer 1 Report

The authors have responded satisfactorily to my comments, so I have no hesitation in publishing the article.

Author Response

Rev#1: The authors have responded satisfactorily to my comments, so I have no hesitation in publishing the article.

Thank you very much for your constructive and encouraging comments; they helped us to improve significantly our work.

Please find attached our comments to this round of revisions

Reviewer 2 Report

I found the changes in the text of the manuscript and the important contribution made by the other reviewers and the addendum in the conclusion very interesting.

Author Response

Rev#2: I found the changes in the text of the manuscript and the important contribution made by the other reviewers and the addendum in the conclusion very interesting.

Thank you very much for your constructive and encouraging comments; they helped us to improve significantly our work.

Please find attached our comments to this round of revisions

Reviewer 3 Report

Dear Authors:

This manuscript has many similar publications on the website.

Maybe all are your further study.

I think so. As the following file is my recommendation:

I thank I can agree to you to publish in this journal.

But you need to change some.

  Thank you.

Author Response

We want to thank the Reviewers for helped us to improve our work; 

Please find attached our comments to this round of revisions

Round 2

Reviewer 3 Report

Dear Authors:

Thank you. I find that.

I can accept your publication.

This manuscript is a resubmission of an earlier submission. The following is a list of the peer review reports and author responses from that submission.

Round 1

Reviewer 1 Report

Scicali et al. performed retrospective observational study evaluating 30 heterozygous familial hypercholesterolemia (HeFH) subjects and 30 healthy controls. They measured plasma lipids at baseline (T0) and after 6 months of treatment with statin plus ezetimibe (T1), and after 6 months of optimized therapy with PCSK9-i (T2), They measured CD34+ cell count at T1 and T2, and PWVs at T0, T1 and T2. At T1 and T2 the CD34+ cell number in HeFH subjects was not significantly different from controls. When HeFH subjects were subdivided into high- and low-CD34+ cell count groups, the CD34+ cell count was significantly different between the two groups at T1 and also compared to controls. They also found PWV to be significantly reduced at T1 and T2 in the HeFH group. Moreover, a significant negative correlation between CD34+ cell number and PWV in the HeFH population as well as in the low-CD34+ HeFH group was found. Based on these results the authors conclude that the PCSK9-inhibitors have a favourable effect the amount of CD34+ cells as well as the PWV values in FH subjects.

The study is interesting and the results are of important value. However, I need to point out some major and minor concerns:

Major concerns

  1. The study is retrospective, which the authors have clearly pointed out, they just need to be more careful when explaining the result of their study. Please tone down the conclusions.
  2. The patient cohort used in this study was very carefully selected from larger group of patients from previous study which needs to be emphasized. The patients selected for the current study did not possess any other risk factor except HeFH which allowed the authors to evaluate the effect of the PCSK9 inhibitors on hyperlipidemia only. (Mandraffino G, Scicali R, Rodríguez-Carrio J, Savarino F, Mamone F, Scuruchi M, Cinquegrani M, Imbalzano E, Di Pino A, Piro S, Rabuazzo AM, Squadrito G, Purrello F, Saitta A. Arterial stiffness improvement after adding on PCSK9 inhibitors or ezetimibe to high-intensity statins in patients with familial hypercholesterolemia: A Two-Lipid Center Real-World Experience. J Clin Lipidol. 2020 Mar-Apr;14(2):231-240. doi: 10.1016/j.jacl.2020.01.015. Epub 2020 Feb 4. PMID: 32111581.)
  3. There is no explanation on why there was the difference in CD34+ cell count within the HeFH group. The only explanation the authors provide is the age of the patients.
  4. Despite that blood pressure of all subjects in this study, i.e. HeFH patients and control group, was normal, the control group showed significantly lower blood pressure, both diastolic and systolic, in comparison to the HeFH group. This might have an influence on PWV. Please add a comment about this.
  5. There is evidence from previous studies that PCSK9 inhibitors decrease Lp(a) levels, but no such data are provided here. Please add a comment on this.

Minor concern

  1. There is 10 out of 35 references that belong to the authors of this manuscript which is over self-citing.
  2. The manuscript would benefit from English editing.

Reviewer 2 Report

I found the study results very interesting as you demonstrated the additional benefits of PCSK9-i in improving vascular homeostasis in this group of patients with HeHF, however, as you said, a randomized controlled trial is required to evaluate the effect of PCSK9-i therapy on these pathways in a large cohort of FH subjects.

Reviewer 3 Report

Dear Authors:

This is a furthermore from the last study.

A single lipid unit and then two units.

However, the wrong data showed.

Not a clinical study results were expressed.

Thus, I think it needs to change too many.

I can not agree with your publication.

Reviewer 4 Report

The manuscript of Scicali et al. is aimed to evaluate the effect of PCSK9 inhibitors (PCSK9-i) on CD34+ CPCs on pulse wave velocity (PWV) in a cohort of 30 heterozygous familial hypercholesterolemia (HeFH) subjects.

Although, the manuscript is well written and the topic could be of interest, it is strongly limited by the study design and major issues need to be reported.

Authors described the study as retrospective, but it is inconsistent with the description of different timepoints related to pharmacological interventions indicated in the section methods [T1 (after 6 months of the maximally tolerated statin plus ezetimibe treatment), and at T2 (after 6 months of PCSK9-I add-on therapy)].

Furthermore, the authors indicated that the determination of CD34+ cells in fresh blood samples was part of a previous prospective study (lines 90-93) (PMID: 32111581) conducted in the same population, but it is not indicated in the section methods of the article above mentioned, formally it is  incorrect.

Authors reported in section results that their study could be considered as a preliminary exploration a potential interrelationship between CD34+ cells number change with PWV change after six months of therapy with PCSK9-i; this could be acceptable if the study was designed as prospective.